

# BTW—Bioinformatics Through Windows: an easy-to-install package to analyze marker gene data

Daniel Morais[1], Luiz F.W. Roesch[2], Marc Redmile-Gordon[3], Fausto G. Santos[4], Petr Baldrian[1], Fernando D. Andreote[5] and Victor S. Pylro[5,6]

[1] Institute of Microbiology of the CAS, Prague, Czech Republic
[2] Centro para Pesquisa Interdisciplinar em Biotecnologia, Universidade Federal do Pampa, São Gabriel, Rio Grande do Sul, Brazil
[3] Natural England, Sheffield, United Kingdom
[4] Biosystems Informatics and Genomics, Instituto René Rachou, Belo Horizonte, Brazil
[5] Soil Department, ''Luiz de Queiroz'' College of Agriculture - ESALQ/USP, Piracicaba, São Paulo, Brazil
[6] Department of Biology, Federal University of Lavras - UFLA, Lavras, Minas Gerais, Brazil

Corresponding author
Victor S. Pylro,
victor.pylro@brmicrobiome.org

## ABSTRACT

Recent advances in Next-Generation Sequencing (NGS) make comparative analyses of the composition and diversity of whole microbial communities possible at a far greater depth than ever before. This brings new challenges, such as an increased dependence on computation to process these huge datasets. The demand on system resources usually requires migrating from Windows to Linux-based operating systems and prior familiarity with command-line interfaces. To overcome this barrier, we developed a fully automated and easy-to-install package as well as a complete, easy-to-follow pipeline for microbial metataxonomic analysis operating in the Windows Subsystem for Linux (WSL)—Bioinformatics Through Windows (BTW). BTW combines several open-access tools for processing marker gene data, including 16S rRNA, bringing the user from raw sequencing reads to diversity-related conclusions. It includes data quality filtering, clustering, taxonomic assignment and further statistical analyses, directly in WSL, avoiding the prior need of migrating from Windows to Linux. BTW is expected to boost the use of NGS amplicon data by facilitating rapid access to a set of bioinformatics tools for Windows users. Moreover, several Linux command line tools became more reachable, which will enhance bioinformatics accessibility to a wider range of researchers and practitioners in the life sciences and medicine. BTW is available in GitHub (https://github.com/vpylro/BTW). The package is freely available for noncommercial users.

## INTRODUCTION

In April 2016, Microsoft announced the release of the Windows Sub-system for Linux (WSL), which is available to Windows 10 users. This distribution consists of a Linux environment compiled through Windows and enables most native command-line tools, utilities and binaries from Linux to run on Windows: the users can now
run Bash scripts and all popular Linux command-line tools like *sed*, *awk*, *grep*, *sort*, *apt*, *ssh* and others. One anticipated outcome was that this effort would bring free software to a wider audience, since Windows is the native Operating System (OS) in ∼85% of the Desktops and Laptops worldwide (per StatCounter for June 2017—http://gs.statcounter.com/os-market-share/desktop/worldwide). However, a further but perhaps unconsidered benefit for expanding bioinformatics accessibility, is that Linux command line tools can be used to run several bioinformatics applications on the hardware available without the need for a dedicated machine, or the hassle of having multiple operating systems on a single machine (dual-boot or virtual machines).

Biologists have recently entered the world of big data (*Marx, 2013*), consisting of cross-referenced databases, ranging from DNA to metabolic pathways (*Cook et al., 2015*). However, it is still challenging to effectively perform data analyses using these databases and to manipulate high throughput sequencing data. This is largely because bioinformatics software is typically developed for Unix Shell (*Seemann, 2013*), and mastery of its command-line interface usually requires intensive training. The command-line interface allows users to easily store and document all the steps taken during the data analysis. This can be achieved through the creation of scripts (chronologically organized list of commands connected by their inputs and outputs) that can be executed thoroughly with one line of code in the command-line interface. This helps to automatize repetitive work and facilitates reproducibility. As previously defined by *Mushegian (2011)*, bioinformatics deals with a dual existence paradigm: as developing technology (the tools), and as a science that applies these tools. In the former, providing a user-friendly graphic interface for data analysis is not always feasible even though command-line gives users and developers flexibility to manipulate and sort data. Attempts have been made to help researchers outside the bioinformatics discipline to use tools dedicated to sequence analysis, for instance, MG-RAST (*Meyer et al., 2008*), SEED2 (*Vetrovsky, Baldrian & Morais, 2018*) and BMP desktop (*Pylro et al., 2016*). Even so, bioinformatics has been the cause of headache for many scientists, even for those who grew up in the computer era.

Making bioinformatics accessible to everyone has been one of the main challenges of contemporary biology. Through the development of bioinformatics tools and training of users in biological data assessment, the Brazilian Microbiome Project (BMP: http://brmicrobiome.org—*Pylro et al., 2014a*) and the Center for Systems Biology (http://c4sys.cz), we observed both students and adept professionals in biological sciences struggle when facing the command-line interfaces of the Unix-based OS (such as Linux and macOSX). The new WSL-Ubuntu feature presented here is aimed to help biologists to access Next-Generation Sequencing (NGS) analysis tools without the above limitations. Although the native Bash tools present at Ubuntu operating systems are useful for manipulating biological data, this distribution does not come without specific bioinformatics packages, which require several steps of settings and installation before usage. To overcome this issue, we have created an easy to follow tutorial to installing WSL (available on http://brmicrobiome.org/tutorialbtw) and a Bash script (freely available on https://github.com/vpylro/BTW) that should be run through the command-line of

WSL-Ubuntu, to set up all the necessary packages for running basic NGS data manipulations and the full microbial community metataxonomic analysis, as previously provided by the BMP to UNIX-based operating systems users (*Pylro et al., 2014b*). We also created a benchmark comparing the performance of the metataxonomic pipeline under the WSL-Ubuntu 16.04 with pure Ubuntu 16.04 distribution and using an Ubuntu as a Virtual Machine inside the Windows 10 platform.

## METHODS

### Application

To demonstrate the functionality and performance of WSL-Ubuntu in running a complete 16S rRNA data analysis pipeline, we assessed the operation of Qiime 1.9 (*Caporaso et al., 2010*), VSEARCH 2.4.4 (*Rognes et al., 2016*) and BMP Scripts (*Pylro et al., 2014b*). This bioinformatics pipeline is used to describe and compare the prokaryotic composition in a group of samples. It relies on the high conservation degree and widespread presence of the 16S gene in prokaryotes. Using high-throughput sequencing technologies to sequence the product of a PCR (polymerase chain reaction), generated with universal primers designed to amplify conserved regions of the 16S gene from a DNA sample, millions of DNA sequences will be generated. After treating the sequences to remove sequencing noise and errors, we quantify and assign taxonomy to every sequence to have an estimate of the whole prokaryotic community in our samples. Further details of the pipeline for metataxonomic analyses and installation of the software are in the Supplementary Material. We found that except for packages that require a graphical display (e.g., core_diversity.py from QIIME), all of them work just as in the pure Ubuntu installation. Programs with graphic output are not yet officially supported by WSL, but from our tests with Xming (http://straightrunning.com/XmingNotes) all the programs performed well. The complete pipeline for 16S rRNA data analysis on WSL is available on http://brmicrobiome.org/win16s (Fig. 1). Briefly, 16S reads data of both forward and reverse amplicons are merged into contigs using the "fastq-join" method (*Aronesty, 2013*) in QIIME. The output file (.fastq) is then quality filtered, trimmed to equal lengths, dereplicated, sorted and binned into operational taxonomic units (OTUs) using VSEARCH commands (*Rognes et al., 2016*). Taxonomy is assigned to each representative sequence using the RDPclassifier (ribosomal database project) (*Wang et al., 2007*) against the GreenGenes (13_8) reference database. An OTU Table (biom format) containing both OTU abundance and taxonomy is constructed using QIIME. Finally, the .biom OTU table is fully compatible with the MicrobiomeAnalyst (*Dhariwal et al., 2017*), a user-friendly web-based platform for microbiome data analyses and visualizations, including taxonomy plots and estimates of $\alpha$- and $\beta$-diversity (http://microbiomeanalyst.ca).

## RESULTS

### Benchmarking

To evaluate the usability of such tool for 16S rRNA amplicon data analysis (Supplemental File 1), we performed a benchmark test using 12,638,185 reads of 16S rRNA gene
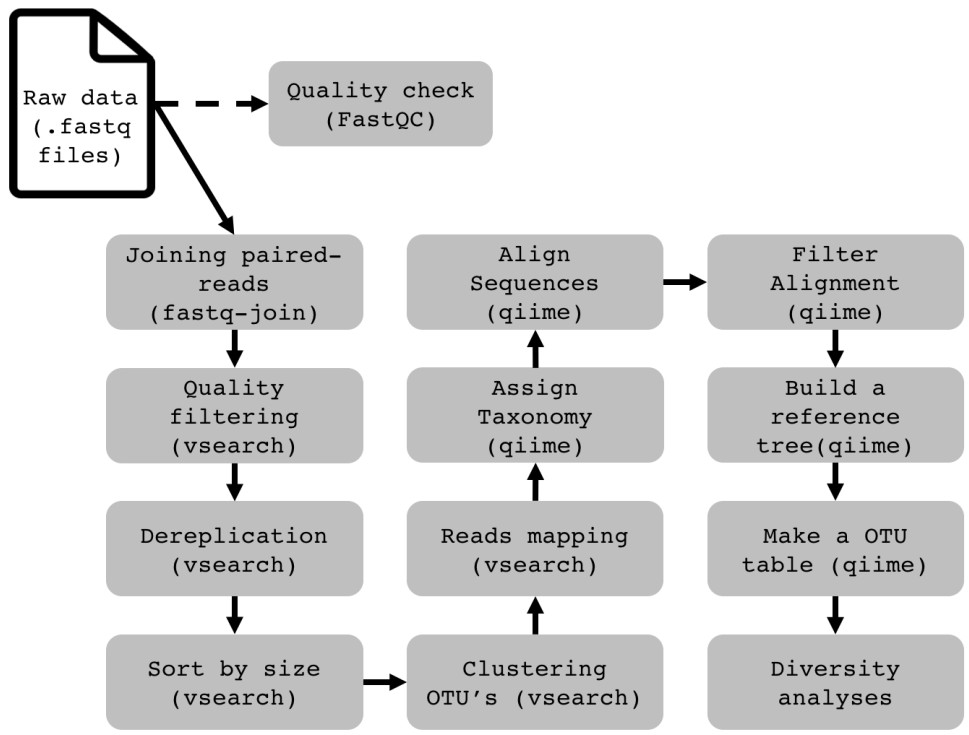

**Figure 1  Flowchart demonstrating the 16S rRNA profiling data analysis pipeline on Windows (WSL).**

data generated from paired-end 150 bp Illumina sequencing (available at https://www.ncbi.nlm.nih.gov/bioproject/241041), in three different scenarios: (1) using the BTW package with the WSL in a Windows 10 machine; (2) using a virtual machine (VirtualBox v. 5.2.6—Oracle) inside a Windows 10 containing Linux Ubuntu 16.04 (BMP OS); and (3) using a dual boot system splitting the hard drive between two operating systems, Windows 10 and Linux Ubuntu 16.04, giving Ubuntu 16.04 full access to processing and RAM memory resources. All tests were performed in a Desktop computer with an Intel® Core™ i7-M640 (two cores; four logical processors—2.8 GHz) and 4 GB of RAM memory. The time elapsed for processing the same data set in each scenario was 3:09 h for the WSL-Ubuntu/BTW, 4:14 h using a virtual machine and 1:27 h in a native Linux Ubuntu 16.04. The outputs were all the same, resulting in a final OTU table in the BIOM format.

## DISCUSSION

Our results show different performances among the tree tested approaches. Virtual Machine took the longest to execute the task, in reason of its nature, requiring complete access to part of the computer's hardware (*Goldberg, 1974*), splitting the processing power by two Operating Systems. The WSL-Ubuntu/BTW was faster than the virtual machine, but still took more than twice the time to execute the task, compared to the plain Ubuntu installation. This is related to two characteristics of the WSL-Ubuntu System: (1) WSL has

to translate various Linux file system operations into the Windows file system structure and to do so, it creates an extra layer of processing to read and write files, making the process slower (the details of this architecture can be read at the Microsoft WSL support blog: https://blogs.msdn.microsoft.com/wsl/2016/06/15/wsl-file-system-support/), and (2) it might be related to the hardware being busy with Windows and Ubuntu basic processes. The plain Ubuntu 16.04 System performed the fastest, without having to share computing resources neither having extra layers for file system manipulation. Although the pure Ubuntu installation is still the fastest approach, this results show the advantage of using WSL-Ubuntu with the BTW installation, in reason of the users having a practical and immediate access to the Unix-style command-line without having to completely change the operating system or suffer from limitations as splitting their hardware or connecting folders from the host and guest operating systems.

Despite being used for 16S rRNA gene NGS data analysis, there is no limitation in using BTW for 18S rRNA and ITS amplicon analyses as well. One concern about the WSL adoption by the scientific community is that this feature is distributed by a commercial company, who may stop supporting it at any time. On the other hand, if developers, scientists and general users show support through uptake, Microsoft may continue to develop and support the feature. In such an event, it is likely that developers of existing tools currently incompatible to WSL will be convinced to join the movement. WSL is still in the beta phase, meaning that some scripts and tools currently used for bioinformatics will not work perfectly. For instance, we experienced fails (segmentation fault (core dumped)) while running QIIME Uclust-based commands and the USEARCH package (*Edgar, 2010*) which seems to be an issue related to the compilation process. To solve the mentioned problem, we adopted similar software to perform the same task, in this case VSEARCH (*Rognes et al., 2016*). Moreover, the bioinformatics users who already learned through this platform, would be able to migrate their knowledge to any full Unix operating system, but without the disadvantage of having to change completely the operating system or use solutions that split their machine's processing power.

## CONCLUSION

BTW has proved useful to facilitate rapid access to bioinformatics resources by Windows users, which will boost analytical capacity for NGS data.

### Funding

This work was supported by the Brazilian Microbiome Project, the National Institute of Science and Technology: Microbiome and the Ministry of Education, Youth and Sports of the Czech Republic grant LM2015055 to Petr Baldrian and Daniel Morais. Victor Satler Pylro receives fellowship from FAPESP (Process 14/50320-4 and 16/02219-8). The funders had no role in study design, data collection and analysis, decision to publish, or preparation of the manuscript.

## Grant Disclosures

The following grant information was disclosed by the authors:
Brazilian Microbiome Project.
The National Institute of Science and Technology: Microbiome: 1.
Youth and Sports of the Czech Republic: LM2015055.
FAPESP: 14/50320-4, 16/02219-8.

## Competing Interests

Marc Redmile-Gordon is employed by Natural England, UK. All other authors declare that they have no competing interests.

## Author Contributions

- Daniel Morais and Victor S. Pylro conceived and designed the experiments, performed the experiments, analyzed the data, contributed reagents/materials/analysis tools, prepared figures and/or tables, authored or reviewed drafts of the paper, approved the final draft.
- Luiz F.W. Roesch and Fernando D. Andreote conceived and designed the experiments, performed the experiments, analyzed the data, contributed reagents/materials/analysis tools, authored or reviewed drafts of the paper, approved the final draft.
- Marc Redmile-Gordon conceived and designed the experiments, contributed reagents/materials/analysis tools, authored or reviewed drafts of the paper, approved the final draft.
- Fausto G. Santos conceived and designed the experiments, performed the experiments, contributed reagents/materials/analysis tools, authored or reviewed drafts of the paper, approved the final draft.
- Petr Baldrian conceived and designed the experiments, analyzed the data, contributed reagents/materials/analysis tools, authored or reviewed drafts of the paper, approved the final draft.

## Data Availability

GitHub: https://github.com/vpylro/BTW.

## Supplemental Information

Supplemental information for this article can be found online at http://dx.doi.org/10.7717/peerj.5299#supplemental-information.

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
