# Peer review of "BTW—Bioinformatics Through Windows: an easy-to-install package to analyze marker gene data"

_PeerJ, doi:10.7717/peerj.5299_

## Round 0.1 · original submission · Major Revisions

Dear authors

As you can see from the comments of the reviewers, your ms needs a thorough revision. Make sure that you improve the ms accordingly as it will be sent to the same reviewers.

Kind regards
Michael Wink
Academic editor

Reviewer 1 ·

Basic reporting

The language used is fine.
The MS is well structured - apart from a clear hypothesis-support/rejection approach.
It should more clearly (in the Abstract) and earlier (in the main text) stated that "bioinformatics" so far only refers to a comparatively narrow window of application.

Experimental design

It is a valuable endeavor to facilitate NGS data analysis for Windows users.
Just one single comparison of performance appears to me insufficent.
As an Ubuntu cluster user I cannot judge the usability for the target Group.

Validity of the findings

The single test outcome seems to be plausible, but see above.

Additional comments

As much as I appreciate the attempt of broadening the application of NGS data analysis, it remains unclear who is intended to read the article. For newcomers to the field, you would have to get into much more detail. Mentioning the actual programs in the figure would not be enough.

Reviewer 2 ·

Basic reporting

see below

Experimental design

see below

Validity of the findings

see below

Additional comments

In this manuscript, the authors introduce a software package which allows users of the Windows operating system to run a variety of bioinformatics analysis. They compare its performance to native Linux-based software and find it to be similar.
I think the manuscript is well written and the presented software has the potential to be useful for a wide audience. I have one major concern though, which is the targeted audience for this paper: If it is aimed mainly at complete bioinformatics and Linux environment beginners, then it lacks a comprehensive introduction of the field of bioinformatics, in particular why it is simply not possible to continue using a standard desktop computer for most high-throughput sequencing analysis. Even people familiar with microsatellite or Sanger sequencing data often think they can just continue with the tools they’re used to. So in my opinion it is important to highlight and illustrate the main feature of these data - hundreds of thousands to millions of reads – which is the reason for why it can’t be analyzed manually or visually (like for example Sanger data).
In addition it would be helpful if the used example (16S rRNA analysis pipeline) would be described / explained in more detail and without jargon. Again, for bioinformatics newcomers, many of the terms and steps might be completely unknown.
People already familiar with bioinformatics might still find the software helpful for teaching purposes, but also then a more extensive introduction is needed.

So in my opinion, to be the intended guide towards bioinformatics, a more comprehensive introduction and illustrative examples should be added to the manuscript.

Detailed line-by-line comments:

Line 30: ‘…is a Bash script and is available on GitHub.’ For people not familiar with the Linux / Bioinformatics environment this sentence might already be too cryptic.

Line 49: ‘… is typically developed for Unix Shell…’ This might be a good starting point to explain why the software is typically developed for Unix shell (and not point-and-click GUI)

Line 52: It is debatable whether a graphic interface is more user-friendly than a command line interface. (Just a comment)

Line 73: ‘Running a complete 16s rRNA data analysis pipeline…’ Please explain this pipeline and also the biological background in more detail, since it might not be familiar to all readers.

Line 82: ‘…using the RDP classifier…’ Please explain abbreviation.

Line 96: ‘…and 32 GB of RAM…’ This amount of RAM is quite above the average of a standard desktop computer, so readers might ask themselves why so much is required.

Line 113: ‘…of having avoided that most daunting first contact barrier…’ To me it is not entirely clear what that barrier is. Is it the need to switch the operating system? Because performing different analysis in BTW still requires using the command line interface right?
-

Reviewer 3 ·

Basic reporting

no comment

Experimental design

The script 'peerj-24722-Illuminabash.sh' should be documented in the same fashion as the
'win_bmp.sh' script provided in GitHub repo.
The same about the installation procedures. A README file containing all the instructions is missing on the github repository.
I strongly suggest to the authors do make a PDF file containing all the instructions available in the website (http://www.brmicrobiome.org/tutorialbtw and http://www.brmicrobiome.org/win16s), and make it available on the package provided in the Github repo, and also including as supplementary material of this manuscript.
This will guarantee that the describe methods are with sufficient detail and information to replicate.

Validity of the findings

-most important issue
On lines 106 to 109, the authors state that:
"Furthermore, WSL is still in the beta phase, meaning that some Bash scripts and
tools currently used for bioinformatics will not work perfectly. For instance, we experienced fails [segmentation fault (core dumped)] while running QIIME Uclust-based commands and the USEARCH package (Edgar, 2010) which seems to be an issue related to the compilation process. "

The authors should propose a possible solution to overcome this potential issue until the final version of WSL be released.
Indeed this will attest the reliability of the BTC pipeline for everyone .

Additional comments

This manuscript describe a new pipeline designed for processing marker gene data, including 16S rRNA.
Although several efforts and tools were developed during the last years for this purpose, this may be the first fully automated pipeline and easy-to-install package as well as a complete pipeline for microbial metagenomic analysis for Windows operational system.
Therefore, this pipeline can be very useful for both students and professionals in biological sciences which generally struggles when facing the command-line interfaces.

---

## Round 0.2 · Minor Revisions

Dear Authors,

Thank you for your resubmission. A minor revision is still required - please see the comment from Reviewer 1.

Greetings,

MWink

Reviewer 1 ·

Basic reporting

-

Experimental design

-

Validity of the findings

-

Additional comments

I see that the authors sufficiently addressed the issues raised by the reviewers.
In the manual, they should not only ask for reference to their work and to the underlying programs, but should also provide the respective appropriate formulations ("Miller AB 2016 Super Program. Bioinformatics 3, 4-5.") including the program version used.

Reviewer 2 ·

Basic reporting

see below

Experimental design

see below

Validity of the findings

see below

Additional comments

The authors have addressed all my comments and concerns to my satisfaction and substantially improved the manuscript. Therefore I can now recommend it for publication.

---

## Round 0.3 · accepted · Accept

Dear Victor

Good news! Your revision is adequate and we are now happy to accept your paper.

Thanks for publishing with us.

Greetings
Michael

# Reviewer 1 ·

Basic reporting

-

Experimental design

-

Validity of the findings

-

Additional comments

thanks for providing details on the underlying programs

Reviewer 2 ·

Basic reporting

see below

Experimental design

see below

Validity of the findings

see below

Additional comments

As previously stated, the authors made all suggested changes to the manuscript. Thus I can now recommend it for publication.

Reviewer 3 ·

Basic reporting

no comment

Experimental design

no comment

Validity of the findings

no comment

Additional comments

I am satisfied with the revised manuscript and response to my first comments. I congratulate them on a valuable contribution.
I just need to point out to the authors a minor comment, to correct the typo on line 139 ("... supporting it at any time., On ...").